# Efflux-Mediated Resistance in *Enterobacteriaceae*: Recent Advances and Ongoing Challenges to Inhibit Bacterial Efflux Pumps

**DOI:** 10.3390/antibiotics14080778

**Published:** 2025-08-01

**Authors:** Florent Rouvier, Jean-Michel Brunel, Jean-Marie Pagès, Julia Vergalli

**Affiliations:** INSERM, SSA, Aix-Marseille University, MCT, Faculty of Pharmacy, 13385 Marseille, France; florent.rouvier@univ-amu.fr (F.R.); jean-michel.brunel@univ-amu.fr (J.-M.B.); julia.vergalli@univ-amu.fr (J.V.)

**Keywords:** accumulation assay, adjuvants and inhibitors, antibiotics, efflux pumps, fluorometry, Gram-negative bacteria, mass spectrometry, multidrug resistance, pharmacochemistry-pharmacomodulation

## Abstract

Efflux is one of the key mechanisms used by Gram-negative bacteria to reduce internal antibiotic concentrations. These active transport systems recognize and expel a wide range of toxic molecules, including antibiotics, thereby contributing to reduced antibiotic susceptibility and allowing the bacteria to acquire additional resistance mechanisms. To date, unlike other resistance mechanisms such as enzymatic modification or target mutations/masking, efflux is challenging to detect and counteract in clinical settings, and no standardized methods are currently available to diagnose or inhibit this mechanism effectively. This review first outlines the structural and functional features of major efflux pumps in Gram-negative bacteria and their role in antibiotic resistance. It then explores various strategies used to curb their activity, with a particular focus on efflux pump inhibitors under development, detailing their structural classes, modes of action, and pharmacological potential. We discuss the main obstacles to their development, including the structural complexity and substrate promiscuity of efflux mechanisms, the limitations of current screening methods, pharmacokinetic and tissue distribution issues, and the risk of off-target toxicity. Overcoming these multifactorial barriers is essential to the rational development of less efflux-prone antibiotics or of efflux pump inhibitors.

## 1. Introduction

Gram-negative bacteria are a serious threat to public health caused by rapidly occurring resistant bacteria to clinically available antibiotics, the development of new broad-range resistance mechanisms, and the scarcity of new antibacterial compounds [1,2,3]. The spread of multidrug-resistant bacterial infections represents a worrying problem for human health in the coming years [1,2,4,5,6]. This challenge is particularly acute with bacteria of the ESKAPEE group, which includes *Escherichia coli*, *Klebsiella pneumoniae*, and *Enterobacter* spp. They show a high and rapid adaptive response to the use of antibiotics, leading to the emergence of multidrug-resistant phenotypes that complicate the treatment of infections and restrict therapeutic options. The resistance mechanisms involve both membranes, the outer and inner membranes (OM and IM), and the activity of the efflux pumps that expel the antibacterial compounds and favor the acquisition of additional mechanisms such as drug target mutation and modifying enzymes that modify and thereby inactivate the antibiotics [5,6,7]. To cross the membrane and access their target, antibiotics must deal with these membrane-associated mechanisms of resistance (MAMs), independent of target mutations or enzyme barriers. These two mechanisms contribute to reducing their internal concentration, thus bringing this concentration below the critical threshold required for the antibacterial action of the molecules.

To date, the majority of studies have focused on identifying active compounds capable of counteracting specific resistance mechanisms, such as target-site mutations or enzymatic modifications [3,4]. In contrast, relatively few investigations have addressed the role of intracellular drug concentration, particularly in relation to antibiotic accumulation. These include limited work on outer membrane (OM) permeability, penetration rates, and the susceptibility of compounds to bacterial efflux systems. Thus, enzyme inhibitors such as clavulanic acid or avibactam for β-lactamases are commonly used clinically in combination with antibiotics, but no therapeutic solution is currently available to circumvent impermeability or efflux pumps [3,4,6,8]. Consequently, with the continuous spreading of multidrug-resistant *Enterobacteriaceae,* there is an urgent need to research and develop original molecules that can overcome the MAMs and restore an internal concentration appropriate to ensure antibacterial effect. At this moment, a serious gap lies in the lack of direct quantification of antibiotics in enterobacterial cells exhibiting membrane impermeability or/and expression of an active efflux pump [4,5]. The respective contributions of the incoming (influx) and outgoing flux (efflux) on the activity of antibiotics are mainly studied by measuring minimal inhibitory concentrations (MICs), rarely by rate-killing assays, and even more rarely by determining the kinetics of accumulation of molecules in bacteria expressing MAMs [4,5,6].

The purpose of this review is to provide a comprehensive overview of efflux pump-mediated resistance in *Enterobacteriaceae* and strategies to overcome it. We first describe the structure, organization, and functional mechanisms of efflux pumps, and their contribution to antibiotic failure. We then explore the recent molecules identified as potential efflux pump inhibitors (EPIs) or disruptors of efflux activity. We discuss current limitations of experimental approaches used to assess efflux inhibition, as well as key pharmacological challenges such as optimal concentration thresholds and tissue distribution. These considerations are essential to better define structure–activity relationships and guide future development of novel compounds suitable for therapeutic use.

## 2. Efflux Pump Structure, Function, and Involvement in Drug Resistance

### 2.1. Structure and Function of Efflux Pumps

Efflux pumps, which are naturally expressed at a basal level in bacteria, display various physiological roles, including adaptation to environmental stresses through osmotic regulation and expulsion of toxic compounds (e.g., bile salts and toxins), as well as bacterial virulence, biofilm formation, and interbacterial communication [9,10]. Among their most critical functions, efflux pumps play a central role in antibiotic resistance, particularly in Gram-negative bacteria.

Bacterial efflux systems are categorized into different major families based on structural features and energy sources. The ATP-binding cassette (ABC) superfamily utilizes ATP hydrolysis to power substrate extrusion. In contrast, the resistance-nodulation-division (RND) family, major facilitator superfamily (MFS), small multidrug resistance (SMR) family, multidrug and toxic compound extrusion (MATE) family, proteobacterial antimicrobial compound efflux (PACE) family, and the p-aminobenzoyl-glutamate transporter (AbgT) family use the electrochemical gradient of protons or sodium ions as their energy source [10,11].

Among these, only members of the ABC, MFS, and RND families have been shown to form tripartite efflux systems that span both membranes in Gram-negative bacteria and are strongly implicated in antibiotic resistance [12]. Therefore, we focus here on these three families due to their structural characterization and clinical relevance. These complexes typically consist of (i) an IM transporter where substrate recognition occurs, (ii) a periplasmic adaptor protein (PAP) that bridges the transporter to (iii) an OM channel forming the exit duct. Representative efflux pumps with available structural data include the RND pumps AcrABZ-TolC, individual components of AcrD and OqxB (which are known to assemble, respectively, with membrane fusion proteins AcrA and OqxA and the outer membrane channel TolC), the MFS pump EmrAB-TolC, and the ABC pump MacAB-TolC (Figure 1). Structural insights—particularly from cryo-electron microscopy and X-ray crystallography—have been instrumental in elucidating efflux mechanisms and informing the rational design of EPIs.

In Gram-negative bacteria, ABC transporters can either function independently, transporting substrates across the inner membrane to the periplasmic space, or act as part of tripartite efflux systems that span both the IM and OM to export substrates outside the cell. The MacAB-TolC system comprises a MacB dimer (inner membrane), hexameric MacA (PAP), and a TolC trimer (outer membrane). The MacB protomer includes a transmembrane domain (TMD) for substrate binding, a nucleotide-binding domain (NBD) for ATP hydrolysis, and a large periplasmic domain. ATP-driven conformational changes in the NBD propagate through the TMD to the periplasmic domain, promoting TolC recruitment and channel opening [13].

Partial data exist for the tripartite EmrAB-TolC system, which exports protonophores like CCCP (carbonyl cyanide m-chlorophenylhydrazone) and nalidixic acid [14]. EmrB appears to interact with cytoplasmic or inner membrane-localized substrates rather than periplasmic ones, unlike RND pumps [15].

The RND pump AcrD, primarily associated with aminoglycoside efflux, also recognizes amphiphilic compounds. Structural analyses reveal that electrostatic interactions are key for substrate recognition, with binding sites located both in the cytoplasm and periplasm, particularly within a central cavity formed by the trimeric assembly [16]. Although AcrD shares structural homology with AcrB, its substrate selectivity may differ, and recent structural data suggest distinct conformational states and possible substrate entry from both the periplasm and the cytoplasm via the central cavity [16].

OqxB is a multidrug RND transporter capable, like AcrB, of expelling a wide variety of substrates. OqxB can be chromosomal or plasmid-borne, which facilitates its spread in *Enterobacteriaceae* [17]. Unlike the asymmetric trimer AcrB, OqxB functions in a symmetric conformation with substrate-binding observed in all three protomers in the tight (T) state. Structural studies suggest the presence of two substrate-binding sites: proximal and distal pockets, as seen in other RND pumps [18].

AcrB remains the prototypical RND transporter for mechanistic/structural/modelling studies. It is a trimeric protein composed of a transmembrane domain (TMD) embedded in the inner membrane, a porter domain (PD), and a funnel domain (FD) extending into the periplasm (Figure 1). The TMD of each monomer includes 12 α-helices; the PD comprises four subdomains (PN1, PN2, PC1, and PC2); and the FD contains two subdomains (PN and PC) [19].

During substrate transport, conformational changes within the TMD are propagated to the porter domain, ultimately driving substrate extrusion [20]. AcrB operates as a functional asymmetric trimer, with each protomer adopting a distinct conformation—namely, access or loose (L), binding or tight (T), and extrusion or open (O)—thereby forming a structural and functional asymmetry crucial for its peristaltic pump mechanism [21,22].

Substrate recognition and transport flexibility in AcrB stem from the presence of multiple substrate entry pathways and two large binding pockets located in the PD: the proximal binding pocket (PBP) in the L protomer, and the distal or deep binding pocket (DBP) in the T protomer. While the PBP is more voluminous in the access state and shrinks in the binding state, the DBP exhibits the inverse behavior [23]. These two pockets are separated by the Phe-617 “switch loop”, a flexible structural element that controls substrate passage from the PBP to the DBP depending on the conformational state of the protomer.

To reach the binding pockets, substrates traverse one of four currently identified access channels (Ch1–Ch4), each with distinct locations, conformational accessibility, and substrate preferences.

Ch1 is located at the interface of the inner membrane and periplasm, accessible in the L protomer. Substrates initially interact with a groove formed by TMD helices 7 and 8, from which they are directed toward the DBP without binding the PBP [24,25]. This channel is used predominantly by low molecular weight drugs such as β-lactams, chloramphenicol, and fusidic acid. Ch2 is found in the periplasm and also accessible in the L protomer. It provides access to the PBP, which serves as a staging area for larger, more complex substrates—such as macrolides—that undergo transition from the PBP to the DBP, induced by structural rearrangements in the PD and movement of the switch loop from the L to T state [24,25]. Ch3 was identified through site-directed mutagenesis. It is accessible in the T protomer and provides direct access to the DBP for planar, aromatic, cationic substrates of low molecular mass [26]. Ch4 is a more recently proposed pathway identified by molecular modeling. It is accessible at the interface between the central cavity and the inner membrane, specifically during the T-to-O transition. It provides an alternative route to the DBP via a groove between TMD helices 1 and 2. This channel appears to be used by carboxylated compounds and molecules containing negatively charged groups [27,28]. β-lactams may utilize both Ch1 and Ch4 depending on their chemical structure and local conditions [28]. This transport is not a passive diffusion through the channels but rather involves a peristaltic-like movement of substrates through a succession of conformational changes, driving them from their initial entry site to the extrusion channel via the binding pockets [24,29]. Importantly, substrates using Ch3 or Ch4 can bypass the PBP and switch loop entirely, highlighting the functional plasticity and complexity of the AcrB transporter.

The active extrusion of drugs by efflux pumps represents a major challenge in the fight against antibiotic resistance due to the versatility of these systems, which arises from their flexible substrate recognition and transport mechanisms. This challenge is further compounded by the inability to generalize transport rules across different efflux pumps or bacterial species [30].

### 2.2. Involvement in Antibiotic Resistance

Efflux pumps play a pivotal role in antibiotic resistance by lowering the intracellular concentration of antimicrobials below effective thresholds.

Efflux-mediated resistance spans across multiple antibiotic classes, including last-resort agents like β-lactams (including novel β-lactam/β-lactamase inhibitor combinations), tigecycline, and eravacycline [31,32]. Efflux systems can act synergistically with β-lactamases to increase resistance levels, particularly in extended spectrum β-lactamase (ESBL)-positive strains, thereby enhancing resistance levels [33].

Efflux alone rarely confers high-level resistance [34], but it contributes significantly to multidrug resistance (MDR) by enabling bacteria to survive sublethal antibiotic concentrations. Indeed, efflux mediated resistance allows the selection and acquisition of additional resistance mechanisms, including target site mutations and horizontal gene transfer [35,36]. Moreover, high efflux activity has been associated with impaired DNA repair systems, increasing mutational rates [36]. Inhibiting efflux can therefore suppress this mutational cascade.

Importantly, the threat of efflux-mediated resistance also comes from its dissemination. For instance, plasmid-borne efflux systems such as tmexCD-toprJ, originally identified in Pseudomonas and now found in Klebsiella pneumoniae, encode RND pumps conferring resistance to multiple drug classes. These plasmids frequently co-carry additional antibiotic resistance genes, amplifying their clinical threat [37,38,39].

Efflux pumps, especially RND transporters, are tightly regulated, and broadly distributed across Gram-negative pathogens, rendering them central targets for novel therapeutics [40,41].

### 2.3. Possible Targets of EPIs

In the context of efflux pump inhibition, several terms are commonly used in the literature to describe molecules that interfere with efflux-mediated resistance. While often used interchangeably, these terms refer to distinct mechanisms or modes of action. In this review, the terms EPI and adjuvant are used in a broad sense, encompassing a range of mechanisms, including direct blockade, functional disruption, and competition. Taking this into account, several methods can be used in order to inhibit the antibiotics efflux by efflux pumps.

EPIs can act through various mechanisms, either demonstrated or hypothesized (Figure 2):-By inhibiting the expression of efflux components (such as AcrA, AcrB, and TolC) or interfering with regulatory pathways [42];-By disrupting the membrane assembly of functional efflux complexes [43,44];-By impairing the energy supply required for active transport (e.g., though PMF dissipation, [36]);-By physically blocking efflux activity through occlusion of the AcrB pockets or TolC channels (e.g., though steric hindrance in the DBP of AcrB [44]);-By inhibiting allosteric transitions involved in the transport cycle [40,45];-By competing for substrate-binding sites [46];-By saturating the efflux capacity through excess substrate [4].

Notably, competitive inhibition requires the EPI to bind with higher affinity than the natural substrate. This binding can prevent the conformational changes needed for the transporter protomers to complete the efflux cycle. Similarly, steric hindrance by EPIs can also impede the conformational cascade of the transport cycle [44]. An EPI may also exhibit substrate-dependent mechanisms of action. For instance, PAβN (phenyl-arginine-β-naphthylamide) competes with substrates for the DBP of RND pumps and interferes with the dynamic conformational cycling of AcrB [47]. Although both PAβN and ciprofloxacin bind the DBP, they interact with different residues; thus, no direct competition occurs. Instead, PAβN slows down the frequency of AcrB conformational transitions through DBP binding [47].

Most EPIs currently under development act by directly binding to efflux proteins, while others have unknown or poorly characterized mechanisms of action [48]. Only a minority of compounds are known to target gene expression or allosteric regulatory sites [40,48].

## 3. Possible Adjuvants’ Structure and Origin

Efflux inhibitor research has been active in both industry and academia since the late 1980s. Early efforts focused on screening compound libraries to find molecules that could boost antibiotics effectiveness by reversing bacterial resistance. In this review, we will focus our attention on the most recent compounds identified as potent efflux inhibitors as many publications have been recently published on the topic.

In the late 1990s, Microcide and Daiichi Pharmaceutical began a joint effort to find compounds that enhanced the fluoroquinolone levofloxacin’s activity against *Pseudomonas aeruginosa*. Screening 200,000 compounds led to the discovery of PAβN [49,50], a dipeptide with little antibacterial activity on its own but capable of reducing levofloxacin’s MIC eightfold. PAβN worked against *P. aeruginosa* strains overexpressing MexAB, MexCD, and MexEF pumps and inhibited efflux in *E. coli* via the AcrAB-TolC system.

PAβN enhanced several antibiotic classes—fluoroquinolones, macrolides, oxazolidinones, chloramphenicol, and rifampicin—but not β-lactams or aminoglycosides. Although optimized versions showed better stability and some in vivo activity, PAβN’s clinical development was halted due to poor pharmacokinetics and toxicity. Still, it remains widely used in lab studies. More recently, PAβN was shown to permeabilize bacterial membranes in a concentration-dependent manner, adding to its effect.

Microbiotix later identified MBX2319 [51], a pyranopyridine compound, through high-throughput screening as an efflux inhibitor targeting the AcrAB-TolC pump in *E. coli*. MBX compounds [44,52] (e.g., MBX3132 and MBX3135) enhanced various antibiotics but were inactive against *P. aeruginosa*. Structural studies suggest they block the drug-binding pocket of AcrB, either by direct competition or by locking the pump in a closed state. Optimized versions, such as MBX3796 [53] and MBX4191 [54], showed better drug-like properties.

Further collaboration led to the discovery of D13-9001 [55], a pyridopyrimidine targeting the MexAB-OprM pump in *P. aeruginosa*. It boosted β-lactam and fluoroquinolone activity in animal models but was limited to MexAB-OprM and was not pursued further.

Other companies also screened for efflux inhibitors in the 2000s. Pharmacia (now Pfizer) identified an arylpiperidine that enhanced novobiocin activity, but further development was limited by toxicity concerns and lack of improvement. Some arylpiperidines also targeted serotonin transporters, increasing safety concerns.

Schuster et al. found the arylpiperazine compound NMP [56,57,58], which significantly boosted levofloxacin and other antibiotics in *E. coli* and *A. baumannii*, but not *P. aeruginosa*. Although promising, its serotonin agonist activity raised toxicity concerns. Later, a more selective analogue ((*R*)-QPE) [59] showed improved safety but was still not suitable for clinical use.

BM-19 [60,61], a fluorescent biphenylmethylene derivative, potentially inhibited AcrAB-TolC and reduced antibiotic MICs, but with moderate success. Structural studies at Shandong University and University of South Australia led to naphthamide-based inhibitors (e.g., A5 and G6) [54,62] that restored antibiotic activity and blocked Nile Red efflux.

A screening campaign at Osaka University identified OU33858 [63], a potential EPI for the MacAB efflux pump in *Salmonella*, which enhanced macrolide activity. Its development status remains unknown.

Taxis Pharmaceuticals, with CARB-X support, is developing efflux inhibitors active against Gram-negative pathogens. One compound, TXA09155 [61,64], showed in vivo efficacy in *P. aeruginosa* infection models and potentiated multiple antibiotic classes.

At Sichuan University, indole derivatives were studied for *E. coli* TolC inhibition. Compounds like 3-amino-6-carboxyl-indolenin showed potential, but no further studies have been reported.

Most known efflux inhibitors are also pump substrates, but recent research suggests allosteric inhibition as a new strategy. At INSERM and University of Lille, fragment screening identified BDM73185 [40], which enhanced various antibiotics without being antibacterial itself. The optimized version, BDM88855 [65,66], showed broad potentiator activity and low toxicity. It binds to a unique site on AcrB’s transmembrane domain, halting the pump cycle.

Further analogues (e.g., BDM91270 and BDM91514) [67] were developed with improved properties. In *A. baumannii*, BDM91531 and BDM91892 [68] enhanced multiple antibiotics, likely by targeting the AdeJ efflux pump. BDM91288 [69] reversed resistance in *K. pneumoniae* and improved levofloxacin efficacy in vivo. Cryo-EM confirmed its binding to the AcrB transmembrane region.

Thus, numerous compounds were found to be active as efflux inhibitors, mostly by direct binding targeting AcrB, AcrA, and TolC specifically or non-specifically, as outlined in Figure 3.

Nevertheless, numerous effective derivatives with promising activity have been identified, although their mechanism of action remains unclear to date. This uncertainty, however, paves the way for new avenues of development.

It is noteworthy that to improve efflux inhibitor screening, a GFP-based reporter assay using the *ramA* promoter was developed [70]. *ramA* encodes a transcriptional activator that upregulates the expression of the AcrAB-TolC efflux system. Its expression is repressed by RamR and can be induced in response to various antibiotics [6,7,12]. Since *ramA* activity increases in response to efflux inhibition, this tool allows high-throughput, cost-effective identification of inhibitors. Screening campaigns using this method identified over 43 potential inhibitors with diverse activity profiles.

## 4. Perspectives

The development and evaluation of EPIs face several scientific and practical challenges, ranging from molecular complexity to pharmacokinetic limitations. These key obstacles (summarized in Table 1) are discussed below to highlight current methodological gaps and future directions.

### 4.1. Efflux Pump Complexity

One of the major challenges in EPI development stems from the versatile substrate selectivity of efflux pumps—particularly RND transporters—which can recognize and extrude a structurally and chemically diverse range of compounds [72,73]. This is compounded by structural plasticity at the access pocket (AP) and distal binding pocket (DBP), as well as the existence of multiple substrate entry channels, which diversify binding routes. Due to this complexity, substrate–EPI interactions cannot be reliably predicted based solely on chemical structure [65]. For instance, a drug may compete with one antibiotic for the DBP but not with others [73]. Even structurally homologous pumps—such as AcrB (from *Escherichia coli*) and AdeB (from *Acinetobacter baumannii*)—can differ significantly in transport mechanisms [30]. Similarly, substrates may bind differently to AcrB and AcrD despite high sequence similarity [74]. These findings highlight the non-transferability of substrate specificity and transport rules across efflux systems or bacterial species.

### 4.2. Limitations of Current Methodologies

EPIs are typically identified via in silico screening or phenotypic assays that measure MIC shifts or dye efflux. However, these methods vary widely in protocols, bacterial strains, and antibiotic pairs used. Such variability complicates the comparison of data across studies and hinders the prediction of in vivo effectiveness. For example, a recent standardized reassessment of 35 reported EPIs revealed that only 17 showed consistent antibiotic-potentiating effects [66]. To improve predictive value, EPI testing should combine complementary approaches, including accumulation assays using clinically relevant antibiotics in MDR clinical isolates. Moreover, because efflux activity and membrane permeability are intertwined [75,76], robust EPI characterization must include analyses of compound penetration and bacterial envelope interactions. A recent publication reviews the different methods currently available, from in cellulo, in vitro, to in silico, to monitor the accumulation and transport of antibiotics in bacteria [4]. The advantages and disadvantages associated with these different approaches as well as the parameters measured in each case are discussed, considering the feasibility and reproducibility as well as the value of the information obtained regarding resistant strains that pose a problem in the clinic [4]. It is clear that a combination of these different methods will provide the necessary data to rationalize the development and use of future molecules targeting this mechanism of resistance to efflux.

### 4.3. Effective Concentration for a Selected Blocker: Assessing Activity with MIC

An important question concerns the concentration of the blocker required to restore drug activity. What is the appropriate index to determine this parameter: MIC or rate-killing measurement, or both?

In this context, how can we assess the antibiotic–blocker ratio, the internal concentration associated with the external concentration required to achieve optimal effect? Importantly, we must also consider the different efflux orthologs and the respective affinities of a given pump for different antibiotic families for the definition of the concentration required to reach optimal effect.

### 4.4. Concerns Related to Tissue Penetration and Pharmacokinetics of EPIs

In addition, another aspect regards the local concentration at the infected site and the diversity of possible sites: lungs, digestive tract, urinary tract, etc., that impose various chemical, physical, and tissue conditions. Moreover, the stability and PK-PD of selected blockers must be determined in the various sites in order to rationalize the used dose and to adapt the blocker–antibiotic ratio to maximize the effect on bacterial efflux activity in various infectious sites.

Finally, the side effects on human organelles must be also studied in order to anticipate possible unexpected actions.

## 5. Challenges for Clinical Applications of EPIs

At this stage, several key aspects must be addressed to provide a rational and comprehensive overview of EPIs and their clinical applications.

Particular attention should be given to: (i) the interaction between the antibiotic and the efflux pump (recognition, binding, and transport), (ii) the interaction between the adjuvant and the efflux transporter (involving the same mechanisms), and (iii) the concentrations required to restore antimicrobial efficacy against resistant bacterial strains.To achieve a meaningful therapeutic effect at a safe dosage (i.e., below toxicity thresholds), the adjuvant must exhibit a higher affinity for the efflux pump components than the antibiotic itself. This relationship can be further explored through modeling and dynamic simulations that incorporate intracellular accumulation data.In all cases, it is essential to determine the effective intracellular concentration of the adjuvant necessary to inhibit efflux system activity—analogous to an enzyme inhibitor constant. The molar ratio between the adjuvant and the antibiotic is also critical, considering the potential for off-target effects on cellular organelles such as mitochondria, or on other transport systems involved in vital cellular processes.Finally, the chemical stability of the adjuvant at the site of infection must be considered to ensure it maintains its efflux-inhibitory activity at the recommended therapeutic concentration.

While EPIs represent a promising therapeutic way, their development faces several persistent obstacles: the structural and functional complexity and versatility of efflux pumps, the lack of universally predictive screening assays, and the intrinsic risk of resistance evolution. Overcoming these challenges will require integrated, multidisciplinary efforts involving structural biology, medicinal chemistry, microbiology, pharmacokinetics and pharmacodynamics, and toxicology of EPIs or combinations of EPI–antibiotic.

Furthermore, to anticipate potential bacterial adaptation or response to this therapeutic strategy, it is essential to evaluate a range of appropriate dosing regimens. This approach allows for the assessment of whether and how resistance mechanisms might emerge against the adjuvant–antibiotic combinations. Recent studies have demonstrated that such resistance development can indeed occur, as observed with β-lactam antibiotics combined with β-lactamase inhibitors [77].

## Figures and Tables

**Figure 1 antibiotics-14-00778-f001:**
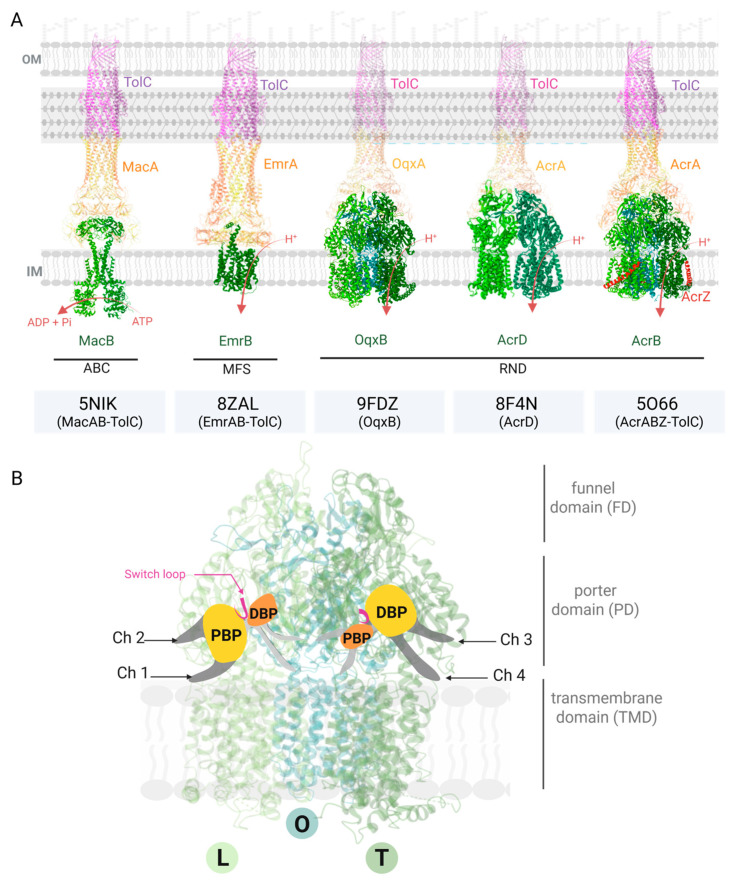
Resolved structures of tripartite efflux pumps involved in antibiotic resistance in *Enterobacteriaceae* and functional cycle of AcrB. (**A**) Representative structures from the ABC (MacAB–TolC), MFS (EmrAB–TolC), and RND (OqxAB–TolC, AcrD–TolC, and AcrABZ–TolC) families are shown. PDB accession numbers are indicated below each structure. (**B**) A close-up of AcrB highlights its binding pockets —proximal binding pocket (PBP) and deep binding pocket (DBP)— substrate channels (Ch1–Ch4), and functional rotation through the loose (L), tight (T), and open (O) conformational states. FD: funnel domain; PD: porter domain; and TMD: transmembrane domain. For OqxAB–TolC and AcrD–TolC, the adaptor proteins and TolC are shown as semi-transparent, as only the inner membrane components (OqxB and AcrD) have available structures in the Protein Data Bank (PDB). Created with BioRender.com. Vergalli, J. (2025); Available at: http://BioRender.com/v194mlo (access on 30 July 2025).

**Figure 2 antibiotics-14-00778-f002:**
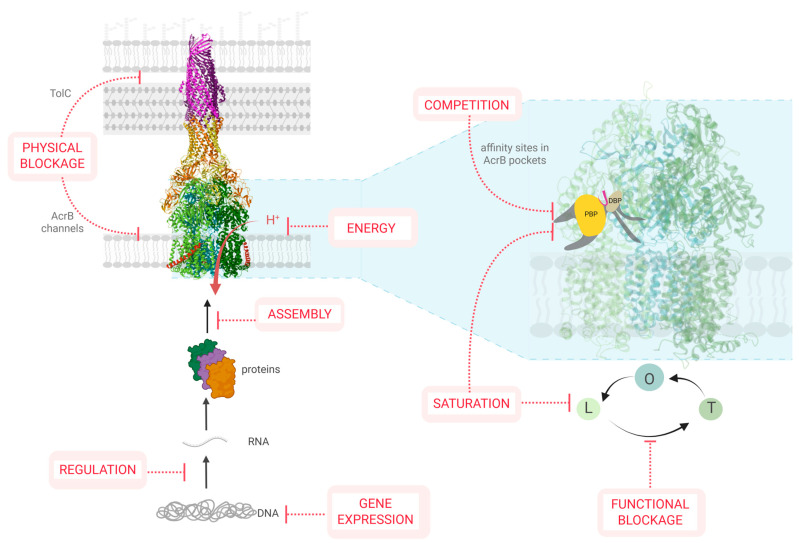
Potential targets of EPIs. This schematic summarizes various molecular strategies by which EPIs can interfere with efflux pump function. These include inhibition of gene expression, regulation, or efflux pump assembly; disruption of the proton motive force; physical obstruction of the exit channels (AcrB, TolC); competitive binding to AcrB recognition pockets (PBP/DBP), saturation of efflux capacities; and functional blockage of the AcrB transport cycle. Some examples and references are mentioned in the Section 2.3. Created in BioRender.com. Vergalli, J. (2025). Available at: http://BioRender.com/1njm3ij (access on 30 July 2025).

**Figure 3 antibiotics-14-00778-f003:**
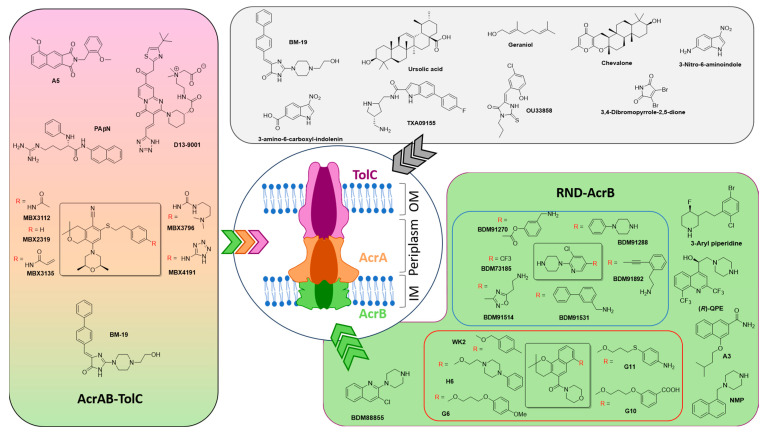
Structural insights into newly identified potent efflux pump inhibitors and their known primary targets: green indicates RND-AcrB, pink-green represents AcrAB-TolC, and grey denotes mechanisms that are not yet clearly understood. The color of the different arrows indicates the target proposed in the pump, respectively.

**Table 1 antibiotics-14-00778-t001:** Key challenges in the development of EPIs.

Challenge	Description/Comment	Impact on Blocking Bacterial Efflux
Efflux pump variability (selectivity for the pumps expressed)	Differences in substrate specificity, structure, and expression levels across species and even homologous pumps. Requires quantification of intra-bacterial concentration of antibiotic [4]. A case-by-case approach is required.	Limits generalizability of inhibitors across efflux pump orthologs.
Substrate diversity and binding complexity (selectivity for a molecule)	Efflux pumps recognize and expel a broad range of structurally diverse compounds using multiple entry and binding sites. Requires dose-effect of kinetics of EPI action on antibiotic concentration inside bacterial cells [4,71].	Predicting substrate–EPI interaction is difficult.
Experimental inconsistency	Protocols, bacterial strains, and antibiotic/EPI combinations vary between studies. Lack of internal standard/control allowing a real comparison between antibiotics, EPIs, bacterial strains, culture conditions, etc.	Hinders reproducibility and comparison of data.
MIC variation and limited readouts	MIC shifts may not reflect the real intracellular drug accumulation or efflux inhibition. Requires accumulation assays, rate-killing assays, and dose effects [4].	Misestimation of the efficacy of EPI.
Dose necessary to control efflux	Difficult to define the concentration needed to impact efflux activity across different pump/antibiotic combinations. Requires bacterial population assay and individual cell assay with a case-by-case approach [4,71].	Influences the timing and drug efficacy.
Diffusion and availability on the internal target	Intracellular accumulation and efflux inhibition may not align with PK-PD. Requires multidisciplinary approaches and the integration of in vitro, in vivo, and in silico results [4]. Requires quantification of EPI and antibiotic inside bacterial cell and analyses of other mechanisms of resistance [4].	Reduces predictive value of in vitro results.
Stability in patient’s body and in infectious site	Different tissues present unique chemical and physiological conditions and drug EPIs may undergo degradation or inactivation in specific biological environments [4].	Decreases the available dose of EPI in the infectious site.
Potential toxicity	EPIs may interact with human transporters.	Complicates clinical use due to side effects.

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
