# Peer review of "Efflux-Mediated Resistance in Enterobacteriaceae: Recent Advances and Ongoing Challenges to Inhibit Bacterial Efflux Pumps"

_antibiotics, 2025, doi:10.3390/antibiotics14080778_

Round 1

Reviewer 1 Report

Comments and Suggestions for Authors

This review focuses on antibiotic resistance mechanisms mediated by bacterial efflux systems. The authors first describe the architecture of efflux pumps and what is known on their mechanisms. They then discuss the various ways to inhibit efflux pumps, the attempts to find inhibitor compounds in the last > 30 years and the major challenges associated with the development of such compounds.

This is a timely review on an important topic. However, based on the abstract I also expected some insight into the techniques available to measure antibiotic concentration in bacterial cells in vivo and to evaluate export efficiency. These issues are touched upon in the text but mainly to explain that these are complex, unsolved questions. I suggest modifying the abstract to better reflect the contents of the review, i.e., by stating that many issues remain in the development of molecules less subject to efflux or of efflux pump inhibitors

Comments

Line 35. Please be more explicit here, as this is the first encounter of these concepts: mutation (of the antibiotic target); enzymes (that modify and thereby inactivate the antibiotic)

Lines 106-107. ‘ABC transporters can function independently’: this statement is not precise enough. Do the authors mean that they can transport some substrates across the cytoplasmic membrane only?

The authors mention seven families of efflux systems. They then describe only three. They need to explain that the other four are poorly characterized, ess prevalent in resistant bacteria and/or less important in antibiotic resistance.

Lines 119-120, it is said that the binding sites of AcrD are in the cytoplasm or the periplasm. In fig 1 and in the text, four channels are described for the homolog AcrB, none of which opens to the cytoplasm. Please clarify.

Lines 102 and 129. Does ’PD’ correspond to periplasmic domain or porter domain?

Line 171-172. A major challenge for what or for whom? Presumably for the fight against antibioresistance, but this needs to be spelled out.

Chapter 2.3. The definitions of the various types of EPIs are confusing as they are not necessarily clear-cut and some partially overlap. Furthermore most of them are not used later in the text. Only the last paragraph, which presents the same concepts in a simplified manner, is useful here.

Lines 317-318. What is the ramA promoter? Which antibiotic(s) activate it, through which regulator? Please provide a reference or explain.

Minor points

Why ‘Gram negative’ Enterobacteriaceae ? All Enterobacteriaceae are Gram negative.

Line 32. ‘complicate their treatment’ should be modified. It is the infections caused by these bacteria that need to be treated.

In the text is mentioned AcrAB -TolC, but an additional component AcrZ is show in the figure 1.

Line 171 (among others): Expel is a verb, not a noun.

Line 203. …antibiotics… (add ‘s’)

Line 108: functional

Line 368: determined

Author Response

This is a timely review on an important topic. However, based on the abstract I also expected some insight into the techniques available to measure antibiotic concentration in bacterial cells in vivo and to evaluate export efficiency. These issues are touched upon in the text but mainly to explain that these are complex, unsolved questions. I suggest modifying the abstract to better reflect the contents of the review, i.e., by stating that many issues remain in the development of molecules less subject to efflux or of efflux pump inhibitors

We thank the reviewer for this insightful comment. We agree that the original abstract did not adequately reflect the scope and emphasis of the review. Therefore, we have edited it to better align with the manuscript’s actual content. The new version now emphasizes the main challenges in the development of efflux inhibitors, the lack of clinically validated methods for measuring efflux efficacy or intracellular antibiotic concentrations, and the gaps that continue to hamper the rational design of EPIs or efflux-evading antibiotics. In addition, the available methods for monitoring antibiotic accumulation have been extensively described/discussed in a recent review on the topic (reference 4), which we cite in the "Limitations of current methodologies" section. We hope this revised version will more accurately conveys the objectives and key messages of the manuscript.

Comments:

Line 35. Please be more explicit here, as this is the first encounter of these concepts: mutation (of the antibiotic target); enzymes (that modify and thereby inactivate the antibiotic)

As suggested, we have revised the sentence in L.35 to more clearly explain the concepts of drug target mutation and enzymatic antibiotic inactivation, to improve clarity for the reader.

Lines 106-107. ‘ABC transporters can function independently’: this statement is not precise enough. Do the authors mean that they can transport some substrates across the cytoplasmic membrane only?

 (L. 82-83): Thank you for pointing this out. We have amended the sentence to clarify that ABC transporters in Gram-negative bacteria can transport substrates across the inner membrane alone or, when part of tripartite systems, across both inner and outer membranes.

The authors mention seven families of efflux systems. They then describe only three. They need to explain that the other four are poorly characterized, ess prevalent in resistant bacteria and/or less important in antibiotic resistance.

Thank you for your observation. We agree that the initial statement lacked clarification. The three families described here are the most directly related to clinical antibiotic resistance, which justifies the attention paid to these three families.

Lines 119-120, it is said that the binding sites of AcrD are in the cytoplasm or the periplasm. In fig 1 and in the text, four channels are described for the homolog AcrB, none of which opens to the cytoplasm. Please clarify.

(L. 93-95): Thank you for this important observation. We agree that AcrB and AcrD do not share identical substrate uptake pathways, despite being homologous RND transporters. AcrB is much better characterized structurally, and the figure focuses on its described substrate entry channels, none of which open directly to the cytoplasm. In contrast, recent cryo-EM studies of AcrD (reference 16, DOI: 10.1128/mbio.03383-22) suggest a distinct architecture, with substrate binding sites located in the central cavity, accessible from both the periplasm and the cytoplasm. We have now clarified this point in the text by explicitly stating that AcrD and AcrB may differ in their transport mechanisms despite structural similarities.

As the structural data for AcrD remain limited compared to AcrB, our figure focuses exclusively on the well-characterized AcrB transporter to illustrate the diversity of substrate entry pathways within RND systems.

Lines 102 and 129. Does ’PD’ correspond to periplasmic domain or porter domain?

Thank you for pointing this out. We confirm that 'PD' refers to the porter domain, which is a specific structural region of AcrB located in the periplasm. We have corrected the inconsistent usage in the text and in the figure to ensure terminological consistency. Although the Porter domain is indeed located in periplasm, the term 'porter domain' corresponds to the structurally and functionally defined nomenclature used in the literature, and not to be confused with the more cellular term 'periplasmic domain'.

Line 171-172. A major challenge for what or for whom? Presumably for the fight against antibioresistance, but this needs to be spelled out.

We agree that the context of the challenge needs to be made explicit. We have now clarified that efflux-mediated drug expulsion represents a major challenge in the fight against antibiotic resistance.

Chapter 2.3. The definitions of the various types of EPIs are confusing as they are not necessarily clear-cut and some partially overlap. Furthermore most of them are not used later in the text. Only the last paragraph, which presents the same concepts in a simplified manner, is useful here.

Thank you for this helpful comment. We agree that the initial listing of definitions (adjuvant, efflux blocker, perturbator, etc.) was unnecessarily detailed. To improve clarity, we have removed this section and instead retained the last paragraph.

Lines 317-318. What is the ramA promoter? Which antibiotic(s) activate it, through which regulator? Please provide a reference or explain.

Thank you for this comment. We have now added a brief explanation of the regulatory role of ramA and its link to the AcrAB-TolC system. We have also cited an appropriate reference (DOI: 10.1128/mBio.01340-20) to support this section.

Minor points

All the minor points are corrected in revised version

Reviewer 2 Report

Comments and Suggestions for Authors

Thank you for submitting this review article focusing on the role of efflux systems in Gram-negative Enterobacteriaceae and the potential strategies to counteract them, particularly through the development of efflux pump inhibitors (EPIs). While the manuscript has substantial strengths in scope and comprehensiveness, it currently requires major revision before it can be considered for publication.

The review shows strong command of the literature and includes recent findings and detailed descriptions of structural and functional aspects of efflux systems. However, several issues related to structure, clarity, depth, and clinical translation limit its accessibility and impact in its current form.

Major Comments:

---- The manuscript would benefit from a clearer and more logical structure. Specifically, I suggest:

Moving the discussion of “experimental challenges,”“PK/PD issues,” and “toxicity” (currently embedded in Section 4) into a dedicated section on translational and clinical challenges (possibly titled “Barriers to Clinical Application of EPIs”).

Rewriting the conclusion to clearly distinguish summary points from future directions, with bullet points or subsections if needed. Currently, the perspectives are somewhat buried in long paragraphs.

Consider inserting a summary table highlighting major EPI classes, known mechanisms, chemical scaffolds, and development status.

---- While the manuscript lists many EPI candidates, it lacks critical discussion on why most have failed to reach clinical use. For example:

Toxicity (e.g., NMP’s serotonin activity) is mentioned but not explored mechanistically.

PAβN’s membrane effects are not contextualized within broader off-target concerns.

You should evaluate whether chemical scaffolds or modes of action correlate with failure or promise in development.

---- The variability of methods used to assess efflux inhibition (e.g., dye accumulation, MIC shift, time-kill, intracellular quantification) is mentioned, but more concrete recommendations are needed. Consider:

Comparing advantages and limitations of each approach.

Proposing a framework for future studies (e.g., standardization of EPI testing in clinical MDR strains with defined efflux profiles).

---- The definitions of efflux modulators, perturbators, blockers, etc., are helpful but dense and potentially confusing. I recommend:

Creating a terminology summary table with mechanisms, definitions, and example compounds.

Being consistent throughout the text in using these terms. In some sections, “EPI” is used broadly, while in others distinctions are made without clarification.

---- Figures :

Figure 2 (mechanism of EPI action) would be greatly improved by annotating representative compounds linked to each mode of action.

Figure 3 lacks sufficient labeling. Please clarify which compounds are shown, and indicate their known or proposed targets in the figure legend or within the figure itself.

---- The current title does not reflect the central focus on efflux pump inhibition as a strategy. Consider revising to something more specific, e.g.:

“ Efflux-Mediated Resistance in Gram-negative Enterobacteriaceae: Recent Advances and Ongoing Challenges in Efflux Pump Inhibitor Development”

---- The manuscript mentions PK/PD and site-specific challenges briefly, but more quantitative discussion is warranted, especially on:

Target concentration ranges for effective efflux inhibition.

Known discrepancies between in vitro potency and in vivo efficacy.

Current gaps in understanding EPI distribution in tissues relevant to MDR infections (e.g., lung, urinary tract).

Minor Comments:

Please proofread the manuscript to correct minor typographical and grammatical inconsistencies (e.g., use of singular/plural, verb tenses).

Ensure consistency in abbreviations and terminology (e.g., “efflux pump inhibitor” vs. “EPI”).

It may help to simplify complex sentences (e.g., lines 373–379) to enhance readability.

Recommendation: Major Revision

In summary, this review covers an important topic and contains valuable material, but

requires substantial revision to meet publication standards. Enhancing structural organization, adding deeper critical analysis, clarifying terminology, and expanding the discussion on clinical translation will significantly improve the manuscript ’ s clarity and impact.

Reviewer 3 Report

Comments and Suggestions for Authors

Comments:

1. This review is well written and successfully demonstrates in-depth discussion on the significance of efflux pump in Gram-negative pathogens.

2. Infographics provided in this article are very informative and relevant to the discussion.

3. The mention of blocking, disrupting, or saturating efflux pumps to restore antibiotic efficacy is insightful and aligns well with current research trends.

4. Challanges and limitations associated with EPIs have been addressed in this review. This could provide a more balanced perspective.

5. The idea of combining efflux pump inhibitors (EPIs) with antibiotics is promising. Could the authors discuss further about this combined therapy ?.

6. Line 208...fonctional assembly appears to be a typo ?

7. Line 221..."interfering with regulatory pathways"  is not clear. which pathways ?

Author Response

Specific comments:

6. Line 208...fonctional assembly appears to be a typo ?

7. Line 221..."interfering with regulatory pathways"  is not clear. which pathways ?

These points have been modified in the revised version (similar points have been mentionned by referee1)

Reviewer 4 Report

Comments and Suggestions for Authors

This review article presents comprehensive review of the current literature on efflux pump inhibitors of Gram-negative bacteria. Mechanism, target site and selectivity among different species have recently been revealed or still remain obscure for these highly attractive molecules. Therefore, current review on this timely and very important issue will attract the attention of researchers from a wide area of interest, thus has particular importance. There are several minor issues that need to be addressed before the manuscript can be accepted for publication.

  1. The font sizes in Figure 1 are particularly small. With the choice of pale colors and semi-transparent figures, some parts become difficult to visualize. Especially Channels 1-4 must be highlighted, as these are the focus of a very important paragraph (Lines 149-170) where different structural units are discussed to prefer one of these gateways.
  2. References must be given to the comments presented in paragraph between the Lines 220-226. Any known examples for the potential target sites of the efflux pump inhibitors must be referenced here.
  3. The paragraph between the Lines 317-321 discusses a newly found GFP-based reporter assay that allows high-throughput examination of potential efflux pump inhibitors. Appropriate references for the development of this reporter and its application on high-throughput studies must be included here.

Author Response

  1. The font sizes in Figure 1 are particularly small. With the choice of pale colors and semi-transparent figures, some parts become difficult to visualize. Especially Channels 1-4 must be highlighted, as these are the focus of a very important paragraph (Lines 149-170) where different structural units are discussed to prefer one of these gateways.

We thank the reviewer for this helpful remark. In response, we have revised Figure 1 to improve its readability. Specifically, we increased the font size throughout the figure, and enlarged the schematic representation of Channels 1–4 to make them more visible. We believe the updated figure now better illustrates the associated text and improves the overall presentation.

  1. References must be given to the comments presented in paragraph between the Lines 220-226. Any known examples for the potential target sites of the efflux pump inhibitors must be referenced here.

Thank you for this insightful suggestion. We have now revised the section to include direct references for demonstrated mechanisms.

  1. The paragraph between the Lines 317-321 discusses a newly found GFP-based reporter assay that allows high-throughput examination of potential efflux pump inhibitors. Appropriate references for the development of this reporter and its application on high-throughput studies must be included here.

We have added the appropriate reference (DOI: 10.1128/mBio.01340-20) to support this section. In addition, other references are now included in the revised version.

Round 2

Reviewer 2 Report

Comments and Suggestions for Authors

After the author's revisions, the manuscript has met the requirements for publication in this journal.